# Regulation of Ion Permeation of the KcsA Channel by Applied Midinfrared Field

**DOI:** 10.3390/ijms24010556

**Published:** 2022-12-29

**Authors:** Yize Wang, Hongguang Wang, Wen Ding, Xiaofei Zhao, Yongdong Li, Chunliang Liu

**Affiliations:** 1Key Laboratory for Physical Electronics and Devices of the Ministry of Education, Xi’an Jiaotong University, Xi’an 710049, China; 2School of Electronic Science and Engineering, Xi’an Jiaotong University, Xi’an 710049, China

**Keywords:** KcsA channel, ion current, permeation mechanism, energy barrier

## Abstract

Ion transport molecules are involved in many physiological and pathological processes and are considered potential targets for cancer treatment. In the large family of ion transport molecules, potassium (K) ion channels, as surface-expressed proteins, show the highest variability and most frequent expression changes in many tumor types. The key to exploring the permeation of K^+^ through potassium channels lies in the conserved sequence TVGYG, which is common in the selectivity filter (SF) region of all potassium channels. We found that the K^+^ flux significantly increased with the help of a specific frequency terahertz electromagnetic wave (51.87 THz) in the KcsA channel using a molecular dynamics combined model through the combined simulation of the constant electric field method and ion imbalance method. This frequency has the strongest absorption peak in the infrared spectrum of -C=O groups in the SF region. With the applied electric field of 51.87 THz, the Y78 residue at the S_1_ site of the SF has a smaller vibration amplitude and a more stable structure, which enables the K^+^ to bind closely with the carbonyl oxygen atoms in the SF and realize ion conduction in a more efficient direct Coulomb knock-on.

## 1. Introduction

In cancer research, new therapeutic targets will facilitate the development of new strategies to combat previously untreatable cancers [1,2]. Changes in the expression levels of ion channels can occur at the genomic, transcriptional, post-translational, or epigenetic levels, and these changes can have a major impact on cellular processes such as cell death, proliferation, migration, and adhesion, including calcium (Ca^2+^), sodium (Na^+^), chloride (Cl^−^) and especially potassium (K^+^) channels [3,4,5,6]. Channel overexpression and dysfunction are strongly associated with tumor progression, invasion, and metastasis [7,8]. Different experimental approaches have demonstrated the relationship between potassium channel blockage and anticancer effects, including induction of apoptosis, inhibition of cell proliferation, and delay of tumor growth [9]. Thus, modulation of K^+^ channel function has been implied as a potential therapeutic modality for different cancers [10,11]. K^+^ channel is a kind of transmembrane protein, which can selectively regulate the permeation of K^+^ inside and outside the cell, and is involved in the formation and generation of electrical signals in excitable cells and non-excitable cells [12]. Regulation of the permeation mechanism will contribute to the role of K^+^ channels in the tumor process. 

K^+^ channel conducts potassium ions (Pauling radius 1.33 Å) at a rate close to the diffusion limit ~10^8^, which has a high transport rate and a high degree of ion selectivity, sodium ions (Pauling radius 0.95 Å) can be reliably excluded at a smaller radius [13,14,15,16,17]. The experimental results showed that although the KcsA channel was derived from prokaryotic cells, it is very similar to the Shaker potassium channel in terms of structure and channel properties. Therefore, the high-resolution prokaryotic KcsA channel provided a direct structural basis for the study of the function and behavior of the eukaryotic potassium channel [18]. The study of KcsA may help us to understand ion permeation in general potassium channels.

The general architecture of KcsA is shown in Figure 1a. The structure is a cross-symmetrical tetramer with four subunits surrounding a central ion channel. The channel can be divided into three regions from intracellular to extracellular: the gate, the cavity, and the selectivity filter (SF) [18,19]. K^+^ arrives at the cavity through the activation gate, enters the narrowest region of the whole channel–the SF region (r = 1.4 Å), in a complete de-hydration, and then it moves to the extracellular and finishes the whole ion transport process [20,21]. All potassium channels have a highly conserved SF with characteristic sequence T-V-G-Y-G [6,17,18]. The SF region consists of four major ion binding sites (S_1_–S_4_), as well as S_0_ and S_cav_ sites above and below the filter. These sites provide tight coordination for fully dehydrated K^+^, and the pattern of ions arrangement at the binding sites ensures both high selectivity for K^+^ and high conductance despite high affinity [17]. 

Under physiological conditions, the prevailing views on the permeation mechanism of K^+^ in the SF fall into two categories: soft knock-on [22,23,24,25] and direct knock-on [19,20,21]. In 2003, MacKinnon and his colleagues proposed that the K^+^ selectivity filter consists of two resident K^+^ separated by a water molecule. The ion pair moves back and forth between the S_1_, S_3_ and S_2_, S_4_ configurations in a concerted manner, denoted as K^+^-W-K^+^-W and W-K^+^-W-K^+^, until a third ion enters the SF region, causing the ion near the exit to leave the channel. Hummer proposed this mechanism as a “soft knock-on” [25]. In 2014, Köpfer et al. proposed a different permeation mechanism—“direct knock-on”. Their work, based on molecular dynamics (MD) simulations, suggested that the key to efficient conductivity of K^+^ lies in the direct Coulomb repulsion between adjacent ions. In direct knock-on, water molecules are excluded from the SF and do not co-permeate with K^+^. So far, which mechanism is dominant still remains controversial [26,27]. Oiki revealed a non-mainstream mechanism—“sparse ion arrays” (e.g., K^+^-W-W-K^+^→W-K^+^-W-W→W-W-K^+^-W→…) and proposed that the ratio of different modes of K^+^ passing through the SF is affected by the concentration of the solution [28,29,30]. 

Although most proteins have a definite conformation, they are not rigid. The flexibility of the conformation caused by the vibration of the atoms in the protein is often critical to its function [30]. Terahertz electromagnetic (THz-EM) stimulus has been reported as an effective way to modulate the physicochemical and dynamic behaviors of biomolecules [31,32,33]. In 2008, Olshevskaya and his colleagues used THz wave output by free-electron laser to irradiate the nerve cells of conidia in still water, and the experimental results showed that THz wave caused the change of membrane potential, the power density of electromagnetic wave was 0.3 mW/cm^2^ [31]. In 2020, X. Liu et al. suggested that a midinfrared stimulation (MIRS) with a specific wavelength of 5.6 µm could effectively enhance the permeation of the K^+^ channel. Their model is a simple molecular model containing only the SF region and uses carbon tubes to replace the phospholipid molecular layer [32]. Therefore, inspired by these works, we infer that there may be a specific frequency that affects the ion transport mode of the KcsA channel in the THz band. 

In 2017, L. G. Cuello and his team resolved the structure of KcsA, which maintains an O/O state of only hundredths of the milliseconds, the selectivity filter is conductive (O), and the pore domain’s activation gate formed by the inner helix bundle is open (O), providing the atomic coordinates of the protein for molecular dynamics simulations [21]. To date, there are two main computational methods for constructing transmembrane potential in molecular simulations: the constant electric field method (CEF) [34,35,36,37,38,39] and the ion imbalance method (IIMB) [34,40,41,42]. The CEF method applies an electric field force to each charged particle to simulate the force of the transmembrane voltage across the plasma membrane. In the IIMB method, the voltage is generated by a small charge difference across the membrane, more closely simulating the uneven distribution of ions on both sides of the membrane in biological systems than in the CEF method. In this paper, we will use the CEF-IIMB combined model to explore the influence of THz-EM on the ion transport of the K^+^ channel. The IIMB method is used to construct the membrane voltage to drive the ion flux, and the CEF method is used to apply an external THz-EM field to the simulation system (Appendix A).

## 2. Results

### 2.1. Ion Current 

When a K^+^ passes through the KcsA potassium channel, it enters SF in the form of complete dehydration and binds to the oxygen atom of -C=O groups at sites S_1_–S_4_ [17] in Figure 2a. Then it passes through each site sequentially and finally reaches the outside of the channel. Therefore, the -C=O groups at the binding sites in the SF region are significant for K^+^ permeation.

Generally speaking, in the infrared absorption spectrum, the absorption frequencies of different functional groups are different. Therefore, according to the simulation data of MD, the infrared absorption spectrum of -C=O groups at sites S_1_–S_4_ in the SF region was calculated by Gromacs software. As shown in Figure 2b, there was a strong absorption fingerprint peak near 51.87 THz in the spectrum. By comparing with the experimental data, the reliability of our calculation results is verified [43,44]. Therefore, we conducted eight independent repeated simulations at the applied electric field frequency at 51.87 THz, with a total simulation duration of 3.2 µs, and obtained the average ion flux (Appendix A) and ion current, as shown in Figure 2c,d. The simulation of the applied field at different frequencies in this paper used the same statistical averaging method as the case of the applied field at 51.87 THz. All the electric field amplitudes of the external THz electric fields were 0.4 V/nm (Appendix A). At the same time, we took the ion current and ion flux obtained in the case without the external electric field, only the IIMB method as a reference, as shown in the chart at the far left in Figure 2c. The results show that the average ion current under 51.87 THz is 1.8 times that of the case without the applied electric field, as shown in Figure 2d. Under the effect of 51.87 THz electric fields, the permeation efficiency of the KcsA channel to K^+^ was significantly improved. This is very close to the 53.53 THz obtained in the experiment [32]. In addition, the effects of THz electric fields of different frequencies on macroscopic effects, such as ion flux and atomic distance of the system, are all more than 100 ns later. Therefore, the statistical calculations of ion flux and ion current are the data after 150 ns of MD simulations. 

Water in the system has strong absorption of THz-EM waves in some frequency bands, so the electric field cannot act on proteins. Therefore, we calculate the absorption spectrum of water in the system. 51.87 THz, with the strongest vibrational frequency of the -C=O group, avoids the absorption of water, as shown in Figure 2b. 

The width of the fingerprint peak was very narrow, less than 2 THz in the infrared spectrum, so we inferred that the -C=O functional groups were more sensitive to the frequency change. In the vibration spectrum of the -C=O group, 50 THz with no absorption peak and 53 THz with small absorption, near 51.87 THz (Figure 2b), were selected as the frequencies of the applied electric field respectively to compare with the case of 51.87 THz field. As shown in Figure 2c, the KcsA channel has the highest average ion flux from 150 to 400 ns under the applied electric field of 51.87 THz (blue, dash line), and the ion flux under the applied electric field at 53 THz (pink, dot line), is lower than that of 51.87 THz. The ion flux without the applied field (black, solid line) and the ion flux with an applied field at 50 THz (green, dash-dot line) are almost close, and the ion fluxes are minimal. Similarly, in the bar chart of Figure 2d, it can be seen more clearly that the current under 51.87 THz is 1.8 times that under 50 THz and without the applied field. In addition, the current at 53 THz field is less than the 51.87 THz case but higher than the current at 50 THz field and the current without the applied field. It is further explained that the KcsA channel’s ability to transport K^+^ is greatly improved under the regulation of the 51.87 THz electric fields.

### 2.2. Ion Permeation Mechanism

K^+^ passed through the KcsA channel’s SF alternatively by the ‘soft knock-on’ mechanism and ‘direct knock-on’ mechanism. The ‘soft knock-on’ mechanism describes the situation in which permeating K^+^ in the SF is separated by water molecules, and the potassium ions in SF come into direct contact during permeation are the situation named ‘direct knock-on’ mechanism. In our simulations, K^+^ permeation events are mainly mixed by these two mechanisms with or without the applied THz fields but mainly conducted by the direct knock-on mechanism (Figure 3b). During the soft knock-on permeation process, K^+^ first occupies the S_1_ and S_3_ sites, and then a new K^+^ appears at the S_4_ site. The K^+^ at the S_1_ and S_3_ sites move to S_0_ and S_2_. The K^+^ at the S_0_ site flows out of the channel quickly. Potassium ions are always separated by water molecules in the SF, and a new K^+^ is not in direct contact with the K^+^ in the SF. In the direct ‘knock-on’ mechanism (Figure 4b), there are 3 K^+^ on the SF sites initially, occupying S_0_, S_2_, and S_3_ sites, and then a new K^+^ enters the S_4_ site. At the same time, the K^+^ at the S_0_ site quickly leaves, and then the remaining 3 K^+^ in the channel are transformed from the ‘S_1_, S_2_, S_4_’ state to the ‘S_0_, S_2_, S_3_’ state. The whole process realizes a complete K^+^ transmission with the direct knock-on mechanism. It can be seen from Figure 4b that the ‘K-K-K-K’ pattern is faster and more efficient than the ‘K-W-K-W’ pattern in the K^+^ transmission of the KcsA channel.

Therefore, we observed and analyzed the ionic configuration states in the SF region under the applied electric field of different frequencies. All cases show that potassium ions are mainly conducted by direct Coulomb knock-on, so we obtained screenshots of six states that may occur during direct ion contacts, as shown in Figure 3a. Moreover, we counted the proportions of these six states and the states that achieved K^+^ conduction by soft knock-on mechanism, as shown in Figure 3b. When the applied field frequency is 50 THz and 53 THz, the conduction states of soft knock-on appear the most, among which it occupies 51.47% in the case of the 53 THz field and 42.9% in the case of the 50 THz field. In the case of the most efficient permeation of 51.87 THz field, the permeation process of soft knock-on only appeared at 7.5%. In the absence of the external THz field, the ion soft knock-on conduction process in the channel accounted for 18.75%. ‘KKKW’ and ‘K0KK’ are the two most populated states, and the probability of these two states appearing with the applied field at 50 THz and 53 THz is significantly less than that with the applied field at 51.87 THz and without the applied electric field.

### 2.3. Potential of Mean Force

We further calculated the potential of mean force (PMF) at the SF, which can be used to characterize the barrier that K^+^ encounters when passing through the channel [41,45]. PMF distribution of SF in the KcsA channel at different frequencies is shown in Figure 5. 

Taking the PMF without the external field as a benchmark (black, solid line), we can see that when the electric field at 51.87 THz is applied to the system, the barrier required for K^+^ to pass through each site is lower than the case without the external field, so K^+^ can pass SF quickly. Under the action of the 53 THz electric fields, although the potential barrier at the S_3_ site is about 1 kT smaller than that without the external field, the barrier required for K^+^ to pass through the S_2_ site to the S_1_ site is higher than others. Under the action of the 50 THz electric fields, the potential barrier of each site is almost close to that without the THz electric field. However, the barrier required for K^+^ to leave the S_2_ site to reach the S_1_ site is higher than that without the external field. In the case of 51.87 THz filed, where the ion flux is the most over the same simulated time, the barrier that K^+^ needs to overcome to pass the SF is lower than that of all sites without the external field.

### 2.4. Pore Radius of SF 

The PMF profile shows a high barrier at the S_1_ site with the presence of the 50 THz filed and 53 THz field and a low barrier at the S_1_ site with the presence of the 51.87 THz field. We analyzed the structure of the channel under the action of electric fields at different frequencies by calculating the average distance over time between the carbonyl oxygen atom on each residue and the carbonyl oxygen atom on the symmetric residue in the SF, as shown in Figure 6. The results of each curve are an average of 8 independent repeated experiments. It can be clearly seen that the change of applied electric field frequency has a great influence on the distance between the symmetric oxygen atoms of Y78 residues at the S_1_ site after 100 ns simulation. It also indicates that the pore radius of the SF region has been changed by the oscillating fields. When the frequency of the applied THz electric field is 50 THz and 53 THz, the radius of the S_1_ site increases, the structure of the SF changes (Appendix A), and K^+^ cannot tightly bind to carbonyl atoms in SF. However, when the frequency of the applied electric field is 51.87 THz, the distance of the oxygen atom at the S_1_ site vibrates at a very small amplitude (Appendix A), and the channel is in the most stable state, which enables the K^+^ to bind closely with the oxygen atom on the SF and realize efficient and fast transmission.

## 3. Discussion

In this paper, we combined the IIMB method and the CEF method, two classical molecular dynamic methods for simulating transmembrane voltage in biological systems, to investigate the effect of electric fields in THz bands on the transmembrane transport of K^+^ in the KcsA channel. By calculating the infrared absorption spectrum of -C=O groups, which are the key binding sites for transporting K^+^ in the SF, we found that under the electric field with a frequency of 51.87 THz, the transfer rate of K^+^ in the KcsA channel is significantly increased, and the current is 1.8 times of that without the THz electric field. This frequency is close to the experimental results by Liu et al. in 2021 [32], who adopted a simplified model, used carbon tubes to simulate the phospholipid layer, and selected only the key atoms in the SF. To further verify that 51.87 THz is a specific resonance frequency, we selected 50 THz with no absorption and 53 THz with weak absorption in the vibration spectrum as comparisons. The ionic current we obtained was as expected, with little change in the ionic current under the electric field at frequency without absorption peak. In addition, under the action of an electric field with a frequency with a weak absorption peak, the ionic current is increased. 

Ion current is a macro statistical result. In order to analyze the internal reasons for the changes in ion flux caused by the applied electric field of a specific frequency, we analyzed the permeation mechanism and ionic configuration states in the SF region. Köpfer et al. suggested that the key to the efficient conductivity of K^+^ lies in the direct Coulomb repulsion between adjacent ions [26,27]. When the applied electric field frequency is 51.87 THz, direct Coulomb knock-on, which is beneficial to high-speed conduction, is the main transmission mode, accounting for 92.5% of the total permeation. When the applied electric field frequency is 50 THz and 53 THz, respectively, the occupation of soft knock-on transmission, which is relatively slow in the ion conduction process, increases greatly. 

Through the analysis of PMF at each SF site, it is clear that under the action of the 51.87 THz electric fields, the potential barrier of each SF site is lower than that without the applied field, and it is more obvious at the S_1_ site. With the applied field of 50 THz or 53 THz, the barrier at the S_1_ site is higher than that in the case without the applied field, while under the action of 51.87 THz electric field, the barrier at the S_1_ site is lower than that in the case of without applied field, and K^+^ can pass through more easily. The main reason for this phenomenon is that the electric field of 51.87 THz makes the Y78 residue at the S_1_ site vibrate less, and the structure of the SF is more stable, which makes the K^+^ and carbonyl oxygen atoms bind closely to achieve fast conduction. 

A common feature of tumor cells is the abnormal expression on the plasma membrane ion channels. This article on how to control the result of the K^+^ permeation mechanism provides a new direction. The applied electric field stabilizes the structure of the K^+^ channel and corrects the reduced K^+^ current. Hence K^+^ channels may become new target molecules for cancer therapy, facilitating the development of new strategies.

## 4. Materials and Methods

The simulated systems selected the KcsA (PDB:5VK6) [21] channel, which has an open inner helix bundle gate and an activated selectivity filter in Figure 1a. This channel was embedded in a patch of a palmitoyl-oleoyl-phosphatidylcholine (POPC) membrane containing 134 lipid molecules, which was then hydrated by 10,359 TIP3P water molecules. KCl (0.3 M) was also added to neutralize the system, resulting in a simulation system of 8 × 8 × 9.4 nm^3^. The system was set up by the CHARMM-GUI webserver [46].

Just as in real cells, transmembrane potentials were established by the IIMB method, and the CEF method provided an external THz-EM field to explore the effect of THz on the ion permeation of KcsA, as shown in Figure 1b.

In addition, the ratio of the magnetic field force to the electric field force of the ions is equal to the ratio of the velocity of the ions to the speed of light. Considering that the velocity of the ions is far less than the speed of light, the influence of the THz magnetic field can be ignored in the molecular dynamic simulation, and only the influence of the THz electric field can be considered. The external THz electric field can be expressed as follow a formula:$$E(t)=E_{0}\cos\left(\omega t+\varphi \right)e_{z}$$

Herein, in our simulation, the electric field strength *E*_0_ was set to 0.4 V/nm, which is the angular frequency, ω and φ denotes its phase set to 0. The electric field is in the Z direction. MD simulations were performed using GROMACS software with CHARMM36 force field [47,48,49]. Periodic boundary conditions were applied in the XYZ direction. PME was used to treat electrostatic interactions exceeding the 1 nm cutoff and set the cutoff of vdW interactions to 1 nm. The LINCS constraint algorithm was used to reset bonds after an unconstrained update of 2 fs. The pressure and temperature were held at 1 atm and 300 K by the semi-isotropic Parrinello-Rahman barostat and the v-rescale thermostat, respectively. The systems performed a steepest descent energy minimization and then equilibrated until the system stabilized. In the 5ns of the NVT and 15 ns of NPT equilibration simulations, the heavy atoms of protein were restrained with force constant of 1000 kJ mol^−1^ nm^−2^ to their starting positions. Lipids, ions, and water were allowed to move freely during equilibration. We duplicated the well-equilibrated single-layered system in the direction of the membrane normal (Z-axis) to prepare a double-layered system for IIMB simulations. In the next production simulations, we set the charge difference on both sides of the cell membrane to 4e to generate the membrane potential.

## 5. Conclusions

In this paper, through the joint simulation of the constant electric field method and the ion imbalance method, it is found that the K^+^ flux is significantly increased with a specific frequency EM wave (51.87 THz) in the KcsA channel. This frequency has the strongest absorption peak in the infrared spectrum of the -C=O groups in the SF region. When the applied electric field was 51.87 THz, the channel mainly transported K^+^ through a more efficient, direct knock-on mechanism. We further analyzed the PMF of the SF region and found that the potential barrier for K^+^ to pass through the channel at this frequency was significantly lower than that of the case without an applied field, especially at the S_1_ site of the Y78 residues. The reason is that the vibration amplitude of the Y78 residues at the S_1_ site is smaller, and the structure is more stable so the K^+^ is closely combined with the carbonyl oxygen atom in the SF region, and the ion conduction is more efficient. There is evidence that voltage-dependent upregulation of K^+^ is associated with cancer characteristics [8]. Different experimental approaches have demonstrated the relationship between potassium channel blockage and anticancer effects, including induction of apoptosis, inhibition of cell proliferation, and delay of tumor growth. They have emerged as key players for new alternatives for cancer diagnosis, prognosis, and therapeutic targets [7,10,11]. Therefore, the modulation of K^+^ channel permeability by electric fields at specific frequencies will likely be an important tool for the treatment of cancer.

## Figures and Tables

**Figure 1 ijms-24-00556-f001:**
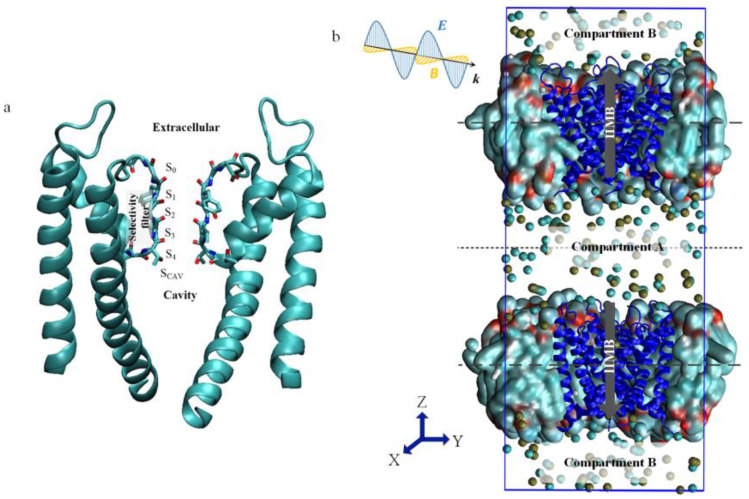
(**a**) Open and conductive state of KcsA with the main K^+^ binding sites of the SF between the four channel subunits. (**b**) The CEF-IIMB combined model. The simulation system consists of two membranes, each including open KcsA, surrounded by water and ions (the yellow balls are K^+^, and the blue balls are Cl^−^). The transmembrane potential is generated by sustaining a small charge imbalance Δq  between compartments A and B by the IIMB method. The CEF method is used to simulate the external THz-EM field.

**Figure 2 ijms-24-00556-f002:**
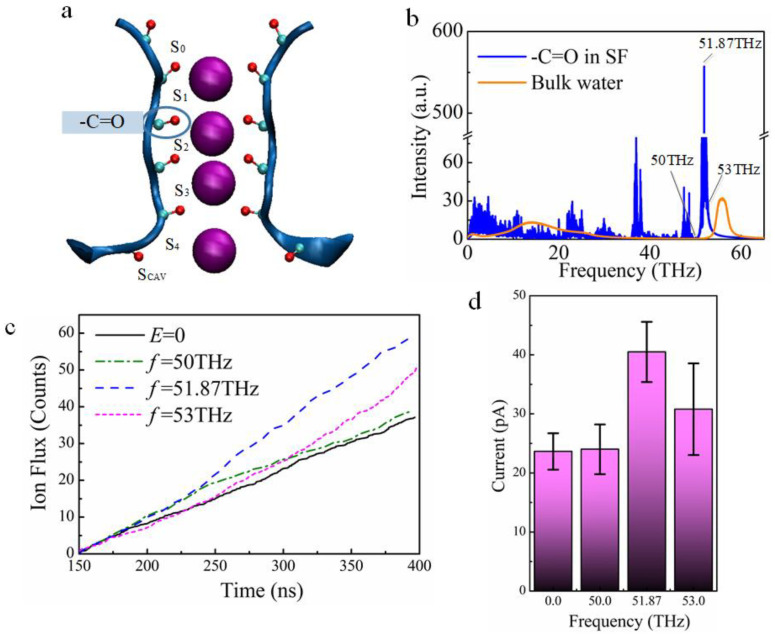
THz electric field increases the ion flux. (**a**) The oxygen atoms in -C=O groups of the SF region are tightly bound to K^+^ (purple spheres are potassium ions). (**b**) Comparison between the absorption of the -C=O groups in SF (blue curve) and that of the bulk water (orange curve). (**c**) Changes of ion flux caused by THz electric field with different frequencies. (**d**) The effect of different frequencies on the current.

**Figure 3 ijms-24-00556-f003:**
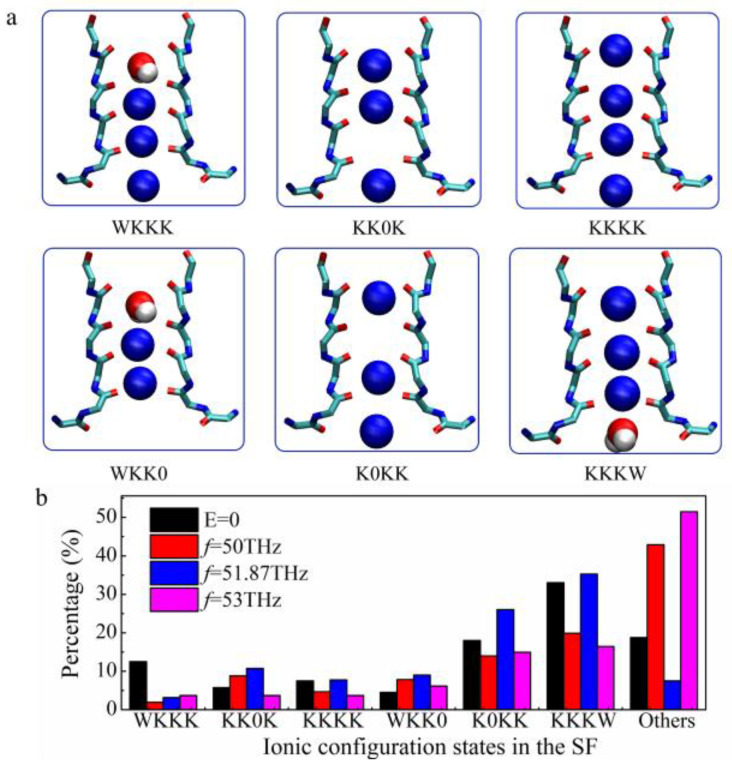
Percentage of ionic configuration states in the SF with and without the applied THz fields at different frequencies. (**a**) SF snapshots of all observed states by direct knock-on mechanism (the blue spheres are K^+^ and the red sphere-white sphere assemblages are water molecules.). (**b**) The distribution of all observed states in direct knock-on mechanism (six states on the left) and soft knock-on mechanism (collectively named Others).

**Figure 4 ijms-24-00556-f004:**
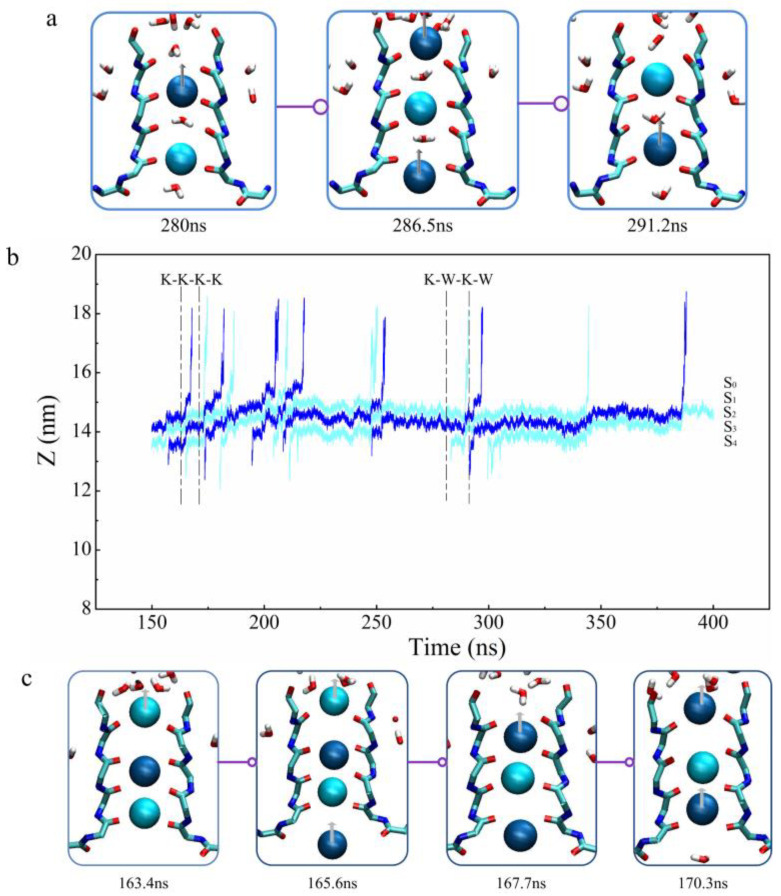
Ion permeation process without the external field randomly selected. (**b**) is a representative trajectory of K^+^ passing through the SF of the KcsA channel in a period of time. (**a**,**c**) select the representative instantaneous screenshots of K^+^ occupying SF sites during the permeation process (We have used spheres of different shades of blue to indicate adjacent K^+^.). (**a**) is a ‘K-W-K-W’ (189–194 ns) pattern and (**b**) is a ‘K-K-K-K’ (288–291 ns) pattern.

**Figure 5 ijms-24-00556-f005:**
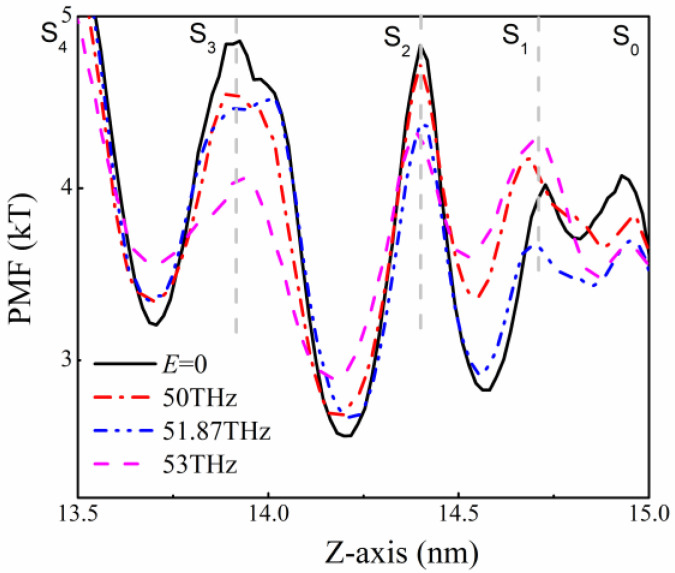
PMF at SF sites with and without the applied electric field at different frequencies, the solid black line is the case without the external field.

**Figure 6 ijms-24-00556-f006:**
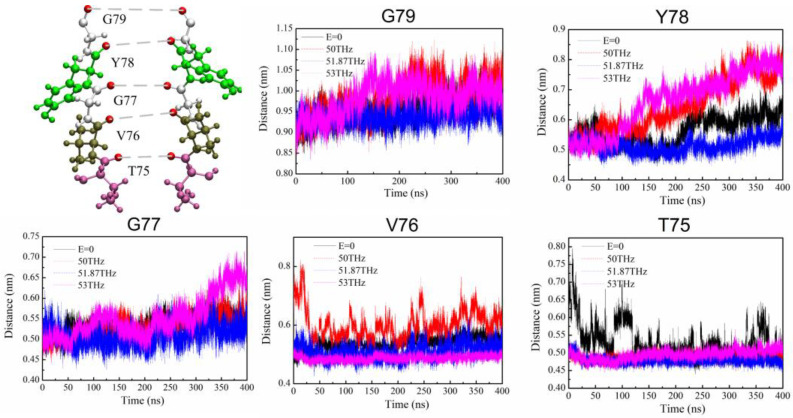
The average distance (*l*) over time between the carbonyl oxygen atom on each residue and the carbonyl oxygen atom on the symmetric residue in the SF. Pore radius (*r*) is equal to *l*/2 minus oxygen atom radius (r_o_).

## Data Availability

The data presented in this study are available in Appendix A.

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
