# Peer review of "Regulation of Ion Permeation of the KcsA Channel by Applied Midinfrared Field"

_ijms, 2022, doi:10.3390/ijms24010556_

Round 1

Reviewer 1 Report

Authors demonstrated that the K+ ion flux is significantly increased with 51.87 THz of EM wave in the KcsA channel. This wave induced the change of  -C=O groups in SF region of KcsA channel, by which K+ ion conduction is more efficient. This modulation protocols in electric field of specific frequency have potential for use in the treatment of K channel-related cancers. 

Overall, this manuscript is a designed study. It will be helpful for readers to read this paper if the introduction explains the molecular mechanism of how intracellular movement of K is related to cancer.

Please write your full name at the beginning of the abbreviation (KcsA channel; potassium channel vs K+ channel).

Reviewer 2 Report

The authors of the article titled” Regulation of ion permeation for the KcsA channel by applied midinfrared field” utilized the simulation of permeation of potassium ions through the selectivity filter of the bacterial Kv channel via the constant electric field and the ion imbalance methods. The authors pointed out that the K+ flux is significantly increased at a particular field EM wave frequency of 51.87THz, supporting the “direct-knock on” mechanism of K+ permeation proposed previously by Köpfer et al., 2014. This specific field frequency stabilizes the interaction of K ions to the carbonyl groups in the selectivity filter region. The simulations did not rule out the “soft-knock on” mechanism, giving a clear demonstration of a mix of these two mechanisms during K+ permeation, as an adaptation of Kv channels at different stimulation conditions. This novel work is exciting and shows details that were not investigated in such depth before. However, there are several issues to be addressed (major changes) before a decision is made to accept this manuscript:

Major changes:

1) Figure 1 lacks labeling and includes the source of the figure in the caption. I prefer to include the S1-S4 region of the SF.

2) The labeling in Figure 2 is unclear.

3) A suggestion: Figures 1 and 2 are to be two panels of one figure.

4) Label Figure 3.a with S1-S4.

5) In the last paragraph of the Discussion, more effort and elaboration should be made by the authors to link the potassium channels with cancer. Kindly, put more details based on the updated literature.

6) Important parts, such as the Funding, Institutional Review Board Statement, Informed Consent Statement, Data Availability Statement, Acknowledgments, and Conflicts of Interest were not specified by the authors.

7) References 39 and 42 are not cited in the text.

8) In lane 287: the citation “Xi et al” is not in the list of References. 

9) Did the authors try to run the simulations using mutate KCsA channel with A78 to see the effect of removing the bulk structure at the R group and investigate if that affects the positioning of the C=O backbone at the SF?        

Minor changes:

- In several places in the text, “K+” to be replaced with “Potassium” when followed by “ions” or only to be written as K+.

-    In Lane 56: remove the hyphen in :con-sists”

-    In lane 133: add a comma before “respectively”

-   In lane 139: Remove “And” at the start of the sentence.

-   In lane 161: remove the word “bar” or replace it with “continuous”

-  In lane 166: add a comma before “such as” and after “system”.

-  In lanes 196-197: Break the big sentence by adding “.” After “mechanism”.

-  In lane 249: remove “be more”.

Round 2

Reviewer 2 Report

The authors addressed all raised questions, comments, and suggestions. Thank you for the great work.